# Spatial-temporal Analysis of Collective Emotional Resonance During Global Health Crisis

## Abstract

The 21st century has already witnessed so many outbreaks with pandemic potential, including SARS (2002), H1N1 (2009), MERS (2012), Ebola (2014), Zika virus (2015), and the COVID-19 pandemic (2019). Using 60 million geotagged Sina Weibo tweets covering over 20 million active accounts, we investigate the collective emotional dynamics on social media in the most recent global pandemic, i.e., COVID'19. This research features two highlights: (1) It focuses on the Chinese population located in the initial epicenter of the pandemic. (2) It examines the initial year after the pandemic outbreak, a critical period where emotions were most intense due to the uncertainty and rapid developments related to the crisis. Using cross-disciplinary methods, we reveal a positive connection between online emotional resonance and geographic proximity, demonstrating a direct mapping between virtual network distances and physical spatial embedding. We propose a percolation-based index to measure the nationwide emotional resonance level with which we illustrate the significant economic impact of the global health issue. Finally, we identify a leader-follower pattern in emotional resonance fluctuations based on time-lag emotion correlations, revealing that less active regions play a crucial role in leading and responding to emotional changes. In the face of long COVID and emerging global health crises, our analysis elucidates how collective emotional resonance evolves, providing potential directions for online opinion interventions during global shocks.

## CCS Concepts

• **Information systems** → *World Wide Web.*

## Keywords

Emotional resonance, Social network, Sentimental analysis, Percolation theory

**ACM Reference Format:**
Anonymous Author(s) . 2024. Spatial-temporal Analysis of Collective Emotional Resonance During Global Health Crisis. In . ACM, New York, NY, USA, 12 pages. https://doi.org/10.1145/nnnnnnn.nnnnnnn

## 1 Introduction

The past five years have survived the long-lasting impacts of one of the newest global health threats in modern history, COVID-19. This pandemic has dramatically altered daily life worldwide, prompting profound changes in how individuals interact, perceive risk, and manage emotions [6, 14]. As the virus rapidly spread across continents, governments implemented stringent measures such as lockdowns, social distancing mandates, and mask-wearing policies to mitigate its transmission. These measures not only aimed at curbing the spread of the virus but also inadvertently reshaped social norms and interpersonal dynamics, triggering a cascade of emotional responses among populations globally [3, 50]. Though human beings have survived the immediate crisis, its effects continue to be felt globally. Indeed, we are experiencing a so-called "post-COVID" era [12]. The global health shock caused by COVID-19 has contributed to the increasing mental health problems rates that may persist for years to come [7].

Sentiment analysis of social media content has emerged as a promising tool for mental health monitoring [35]. As individuals navigate unprecedented challenges and events, they spontaneously share their emotional experiences with others [26, 34]. Emotional resonance occurs as individuals resonate with the collective emotional experiences of their communities and the broader global population [33]. The recent COVID-19 crisis has brought emotional resonance to a new dimension as many individuals have turned to virtual platforms such as social media, video calls, and online forums to connect with others due to the lockdown[29]. Online social platforms such as Facebook, Twitter, Instagram, and TikTok have reported spikes in daily active users, content creation, and interactions since the onset of the pandemic [37, 40]. In this case, social media platforms have become integral spaces for individuals to express their thoughts, feelings, and experiences. An increasing number of studies have underscored the role of social media platforms as virtual support networks, avenues for seeking information, and outlets for emotional expression and connection in physical distancing measures [5, 31]. This characteristic, coupled with the unparalleled access to real-time, large-scale data provided by social media platforms, renders social media data a valuable resource for exploring the nuanced interplay between individual emotions, societal dynamics, and broader contextual factors during global health crises [28, 43].

Despite the increasing interest in the psychology and mental health of individuals under external shock, there remains a notable lack of in-depth investigation into emotional resonance during the global health crisis. To this end, this paper conducts cross-disciplinary research on online collective emotional resonance triggered by the most recent global pandemic, COVID-19. Though five years have passed, this pandemic is the only major global health crisis with the most extensive and diverse social media dataset available for analysis [9]. Our research featured two key points: (1) Focus on the Chinese Population: China was the first country affected by the virus, making it a critical region for understanding the initial emotional and social responses to the pandemic [17]. (2)

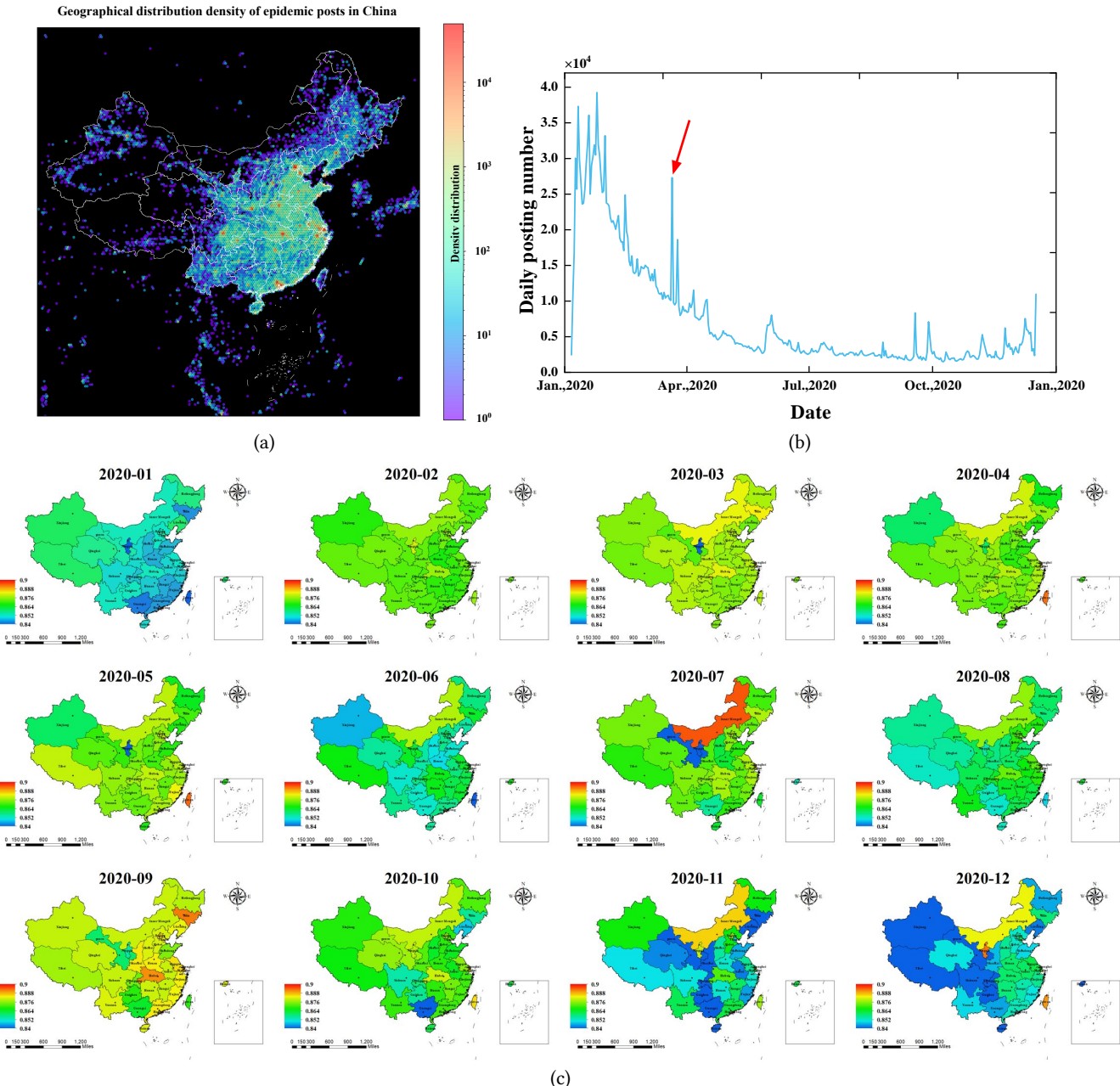

**Figure 1: The spatial and temporal evolution of the Weibo tweets and corresponding expressed emotion in China. (a) The spatial distribution of Weibo post density shows that Wuhan (the center of global rhetoric) and Beijing (the capital city of China) are two of the most active posting cities. (b) The temporal evolution of the pandemic-related tweets number. The red arrow indicates a tweeting peak concurrent with the Qingmin Festival. (c) The temporal evolution of the spatial distribution of the average expressed emotion strength throughout the year 2020 indicates apparent spatial and temporal heterogeneity.**

Focus on the Initial Year: This timeline recorded the most immediate and raw emotional responses as people faced the uncertain evolution of the global health crisis.

A Chinese-based dataset [16] comprises 60 million tweets on Sina Weibo (the Chinese equivalent of Twitter) from December

1, 2019, to December 31, 2020, is used to navigate the emotional resonance throughout the first year of the pandemic global spreading. This dataset includes 20 million active users. To concentrate on the pandemic-related tweets, a keyword screening method involving terms like 'COVID-19' and 'coronavirus' was employed

[16]. Figure 1 (a) visualizes the spatial distribution of the number of related tweets over the year 2020. Regions with big cities (such as Beijing, Wuhan, and Shanghai) tweet more frequently in response to the pandemic. Figure 1 (b) presents the temporal evolution of the number of pandemic-related tweets. Except for peaks at the early stage of the pandemic breakout, another clear peak in tweeting activity occurred on April 4th, 2020 (indicated by the red arrow), coinciding with the traditional Qingming Festival in China, a day when people honor their ancestors and deceased relatives. The sentiment analysis of the tweets is conducted using the state-of-the-art Sentiment Knowledge Enhanced Pre-training (SKEP) model [38], which quantifies both positive and negative expressed emotions of the tweets (see Methods for details). To focus on the strength of emotion, we take the absolute value of the quantified emotion score. Figure 1 (c) illustrates the average sentiment score of each province in China in 2020. These scores are normalized between 0 and 1, with a higher score closer to 1 indicating a relatively strong expressed emotion (either positive or negative). In comparison, a lower score suggests a relatively weak emotion.

Utilizing the geotagged social media data and our measures of expressed emotion, our study investigates the collective emotional resonance on social media platforms focused on China during the most recent global health crisis. We first propose the concept of emotional resonance based on the Pearson correlation coefficient and explore the spatial connection among expressed emotions. Then, we develop a percolation-based indicator for nationwide resonance measurement by constructing the emotional resonance network. To further investigate the temporal dimension of emotional resonance, we introduce a time-lag emotional correlation coefficient and identify the emerging patterns in emotional resonance fluctuations.

## 2 RELATED WORK

In recent years, the study of emotional dynamics using data from social media has gained increasing attention. The COVID-19 pandemic has offered a unique context to explore how collective emotions evolve over time and space, influenced by external shocks and crises.

Existing studies have investigated emotional expression patterns using sentiment analysis tools in various contexts. Bollen et al. [4] pioneered the use of Twitter data to gauge the collective mood of a population, providing an early framework for analyzing how emotions spread on social networks. Their study showed that large-scale societal events, such as financial markets or political changes, directly affect collective emotions, which can be observed through social media activity. With the onset of COVID-19, increasing attention has been devoted to understanding the emotional impact of the pandemic. For instance, Abd-Alrazaq et al. [1] conducted a large-scale sentiment analysis on Twitter, revealing that public sentiment exhibited clear patterns of anxiety and concern as the pandemic evolved globally. Similarly, Li et al. [23] analyzed emotional expression on Sina Weibo during the early outbreak of COVID-19, finding distinct peaks of negative sentiment coinciding with key events, such as the announcement of lockdowns. Several surveys have addressed this topic with similar findings [2, 27, 39]. These results emphasize the great importance of exploring public

sentiment via social media text, especially in the context of external shocks.

In the context of sentiment analysis, there has also been an increasing interest in developing models that capture the dynamics of emotional contagion—how emotions spread from one individual to another across social networks. The infectious disease model has traditionally served as an esteemed method for investigating the mechanisms underlying emotional contagion [41]. Among their exploration, various emotional infection models have been proposed based on SIS, SIR, and their variations, such as the personalized virtual and physical cyberspace-based emotional contagion model (PVP-ECM) [15], the stochastic event-based emotional contagion model (SEEC) [36], and the dynamic multiple negative emotional susceptible-forwarding-immune model (MNE-SFI) [48], etc. To accurately represent the actual state of nodes within social networks, researchers began utilizing complex network theory. For example, Zhu et al. [52] defined the internodal contagion probability based on the network structure and constructed the SIpInRS model for netizen emotion contagion. In another study, Wang et al. [42] introduced multilayer networks to study investor sentiment and stock return connectedness. Meanwhile, Xie et al. [49] and Lu et al. [24] explored the emotional spreading phenomenon, highlighting how emotion contagion can be modeled as a phase transition process like the percolation process.

Based on the results of sentimental analysis and emotion detection, researchers further explore the spatial and temporal properties of online emotional expressions from different aspects. For example, Wang et al. [44] studied the global emotional impacts of the pandemic and proved that COVID-19 outbreaks caused steep declines in expressed sentiment globally. Jabalameli et al. [18] analyzed COVID-19-related social media data in the U.S., detecting public opinion and sentiments related to vaccination and mapping their spatial and temporal distributions. Zhou et al. [51] proposed a spatial-based pandemic-cognition-sentiment (PCS) conceptual model and revealed that the pandemic has a depressive effect on public sentiment in the center of the outbreak. Ding et al. [10] explored the spatiotemporal distribution patterns of negative emotions in mainland China during different stages of the COVID-19 pandemic and indicated that the pandemic significantly intensified the clustering effect of negative emotions. These studies highlight the potential of sentiment analysis to reveal how public emotions fluctuate across different geographic regions and periods, demonstrating the significant impact of the pandemic on online emotional expression.

## 3 Methods

### 3.1 Data processing

The dataset used in this paper is described in [16]. The dataset is processed by applying a predefined list of topic-related keywords to filter out tweets related to the targeted event. We utilize each province and municipality's latitude and longitude coordinates, dividing them into 34 grid points for clustering operations. Data points falling within each province's coordinates range are considered representative of that province. After matching and filtering geographic coordinates, we have a dataset comprising 2.55 million posts and 0.755 million related users.

The Sentiment Knowledge Enhanced Pre-training (SKEP) model, an emotion-enhanced pre-training algorithm developed by Baidu Research, is employed for calculating the sentiment score of the tweets. If a statement is positive, the model assigns it a positive emotion score; if it is negative, it assigns a negative emotion score. Using the language model described above, we obtain monthly average expressed emotion values for 34 provinces and municipalities in China. Each province and municipality generates multiple emotional values daily. We convert negative emotional values to their absolute values to gauge overall emotional resona. Next, we calculate each province's average daily emotional value by averaging these values. The average accuracy of the SKEP model for Weibo-based emotion prediction is approximately 86% (the accuracy is obtained by averaging the model accuracy over three datasets, which can be found in Appendix Table 1).

## 3.2 Emotional resonance coefficient

The expressed emotions in response to the pandemic show obvious spatial heterogeneity, as shown in Figure 1 (c). To explore this spatial distribution property, we develop the emotional resonance coefficient (ERC) to measure the degree of emotional resonance between different regions. ERC is defined as the Pearson correlation coefficient of the average sentiment score between two provinces $x$ and $y$, $R^p_{xy}$, as formulated by Eq. (1),

$$R^p_{xy} = \frac{\sum_{t=1}^{T} x_t y_t - \frac{1}{T}(\sum_{t=1}^{T} x_t)(\sum_{t=1}^{T} y_t)}{\sqrt{[\sum_{t=1}^{T} x_t^2 - \frac{1}{T}(\sum_{t=1}^{T} x_t)^2]}\sqrt{[\sum_{t=1}^{T} y_t^2 - \frac{1}{T}(\sum_{t=1}^{T} y_t)^2]}},$$
(1)

where $T$ is the number of days each month over which the monthly emotional resonance coefficient is calculated. $x_t$ and $y_t$ are the expressed emotion scores of the two provinces on the $t$-th day, respectively. The ERC ranges from -1 to 1, where a positive value indicates a strong positive correlation between the expressed emotion of the two provinces, and a negative ERC signifies a negative correlation.

In our analysis, we focus exclusively on positive ERC values. Negative ERC values are excluded as they hold minimal relevance to emotional resonance, which, according to its definition, tends to describe positive correlations and interactions of expressed emotion.

## 3.3 Country-wide emotional resonance index

Percolation theory provides a natural but powerful description of spreading dynamics on networks. Recent years have witnessed emerging results in adopting the percolation theory in the analysis of social networks [46, 47]. Here, based on the proposed concept of emotional resonance, we construct the emotional resonance network (ERN) and apply percolation-based analysis to measure the country-wide emotional resonance level.

The network consists of 34 province nodes, with links between each pair of nodes weighted by the Emotional Resonance Coefficient (ERC) of the province pair. Figure 2 (a) illustrates the emotional resonance network for February 2020, which is a fully connected network. An initial observation reveals that users located in the middle and southeastern regions of China, which encompass relatively

developed cities, exhibit stronger positive emotional resonance with each other. In contrast, those in the northwestern regions present weaker emotional resonance. For better illustration, we present a less dense network consisting of 50 randomly selected links. A similar observation can be drawn from Figure 2 (b).

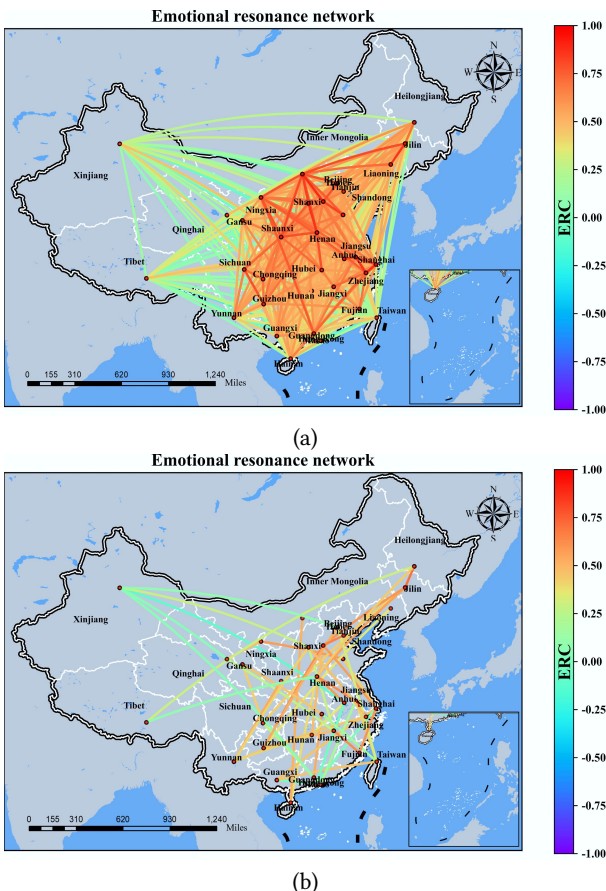

Figure 2: (a) The complete emotional resonance network (ERN) for February 2020. (b) A less dense network with randomly selected links for better illustration.

Then, the country-wide emotional resonance is modeled by a link percolation process. For each correlation link in the network, a tunable percolation parameter $q$ is defined to determine the state of the link. The state of each correlation link $e_{ij}$ with the emotional resonance coefficient $R^p_{xy}$ will be classed into two cases: active state for $R^p_{xy} \geq q$ and inactive state for $R^p_{xy} < q$, i.e.

$$e_{ij} = \begin{cases} 1, & \text{if } R^p_{xy} \geq q, \\ 0, & \text{if } R^p_{xy} < q. \end{cases}$$
(2)

We remove all links with inactive states and calculate the connected clusters in the rest of the network. A connected cluster indicates that nodes (provinces) within it are inter-correlated with a relatively higher emotional resonance level larger than $q$. We then tune $q$ from 0 to 1 with an interval $\Delta q$=0.001, representing

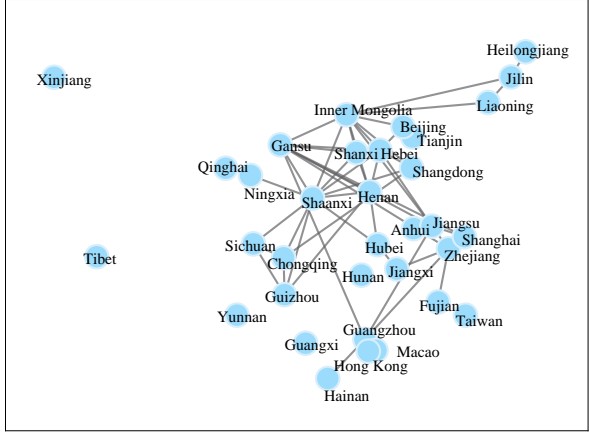

(a)

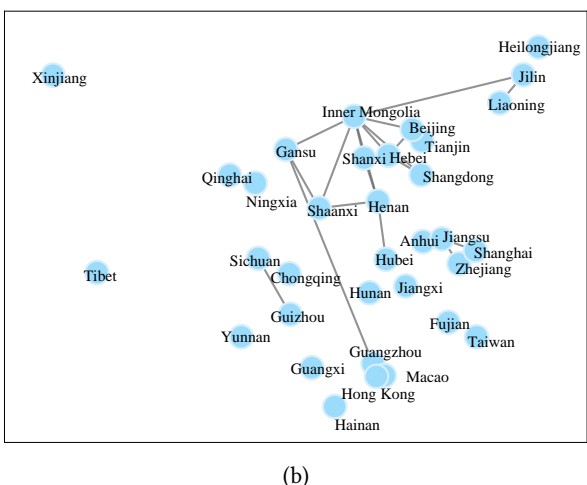

(b)

**Figure 3: The disintegration of an emotional resonance network. The emotional resonance network (a) keeps globally connected when $q$=0.66, and (b) decomposes when $q$=0.76.**

an increasing filter of the emotional resonance level. With the increasing $q$, a higher proportion of links is removed. This leads to a decrease in the size of the giant component of the network, $G$. According to the percolation theory [22], the second-largest component, $SG$, increases and reaches its maximum at a critical parameter $q_c$, signifying the network undergoes a phase transition from global connectivity to disintegration. Figure 3 (a) and (b) show an example of the network before and after the decomposition. The threshold, $q_c$, is obtained by identifying this critical transition. Here, we redefine $q_c$ as an indicator to measure the level of country-wide emotional resonance, i.e., the emotional resonance index (ERI).

### 3.4 Time-delayed cross-correlation

To explore the fluctuations of emotional resonance with time, we define the time-delayed cross-correlation [11] of emotional resonance. The fluctuation of emotional resonance $T_i(t)$ is obtained by subtracting the original expressed emotion scores of province $i$ by

its mean value over each month. For each pair of provinces $i$ and $j$, the time-delayed cross-correlation between the time series ($T_i$ and $T_j$) of each month is calculated by,

$$C_{ij}^{(t)}(\tau) = \frac{\langle T_i(t)T_j(t-\tau)\rangle - \langle T_i(t)\rangle\langle T_j(t-\tau)\rangle}{\sigma_{T_i(t)}\sigma_{T_j(t-\tau)}}, \quad (3)$$

where $\sigma_{T_i(t)}$ is the standard deviation of $T_i(t)$, $\tau \in [-\tau_{max}, \tau_{max}]$ is the time lag, with $\tau_{max}$ = 6 days (one-week interval), $t$ is the starting date of the time series. $\langle\cdot\rangle$ is the average operator. We locate the largest absolute value of $C_{ij}^{(t)}(\tau)$ and denote the corresponding time lag as $\tau_{c,ij}^{(t)}$.

Scientific studies on collective behavior have confirmed the fundamental rules of "following" and "attraction" within group movements[32]. Here, we incorporate these rules into our study of collective emotional resonance. The positive sign of $\tau_{c,ij}^{(t)}$ indicates that the emotional resonance fluctuation of $j$ lags behind $i$ (i.e., $i$ leads $j$), while a negative sign of $\tau_{c,ij}^{(t)}$ signifies $i$ follows the fluctuation of $j$. We define the strength of this leader-follower relationship by calculating the normalized time-delayed cross-correlation, $S_{ij}^{(t)}$,

$$S_{ij}^{(t)} = \frac{C_{ij}^{(t)}(\tau_{c,ij}^{(t)}) - \langle C_{ij}^{(t)}(\tau)\rangle}{\sigma_{C_{ij}^{(t)}(\tau)}}. \quad (4)$$

Ideally, the larger the value $S_{ij}^{(t)}$ is, the stronger the leader-follower relationship between $i$ and $j$.

Then, we construct a complete network based on $S_{ij}^{(t)}$. The network consists of 34 province nodes, where the direction of the link is determined by the sign of $\tau_{c,ij}^{(t)}$, and the weight on the link is $S_{ij}^{(t)}$. The weighted indegree and outdegree of each node is defined as,

$$OD_i^{(t)} = \sum_{j\in N, j\neq i} S_{ij}^{(t)}(\tau_{c,ij}^{(t)} > 0), \quad (5)$$

$$ID_i^{(t)} = \sum_{j\in N, j\neq i} S_{ij}^{(t)}(\tau_{c,ij}^{(t)} < 0), \quad (6)$$

where $N$ is 34. Here, larger weighted outdegree values reflect a stronger leader effect of the node, while larger indegree indicates a stronger follower effect in terms of emotional resonance fluctuation.

## 4 Results

### 4.1 The spatial dimension of emotional resonance

We examine ERC values across China's 34 provincial administrative regions. The monthly ERC value of each province pair is shown in Figure 4 (a). A quantified spatial heterogeneity of emotional resonance in response to the pandemic can be observed, with certain regions exhibiting stronger correlations in expressed emotions than others, e.g., Hubei and Sichuan in January. Notably, January displays the strongest (red) emotional resonance among all periods, likely due to it being the first month following the outbreak of the pandemic. As a pivotal period marked by the initial shock and upheaval caused by the pandemic's emergence, individuals and communities

may have experienced heightened emotional responses and shared sentiments.

This spatial heterogeneity prompts further investigation into the underlying factors shaping the emotional resonance dynamics across different geographic areas. To this end, we explore the ERC as a function of physical distance $d$ between each pair of provinces, revealing a negative linear correlation between the two parameters, i.e.,

$$ERC = -k \times d + \beta, \tag{7}$$

where $k$ is the fitting slope obtained by the least squares fitting method [13]. The physical distance $d$ between province pairs is determined by calculating the Euclidean distances between their geographic centers. We also find that, despite temporal variations of the ERC value between each province pair, the slope $k$ tends to be stable ($k = 0.05\pm0.00579$) for all observed months. These high-quality negative correlations found here in different province pairs and different periods highly suggest that the emotional resonance defined here may reflect an intrinsic property of online expressed emotion independent of the geographic cultures that change from province to province.

This result indicates that populations in regions with closer physical proximity tend to exhibit stronger emotional resonance on social media, while those farther apart display weaker correlations in expressed emotions. Considering the relatively low population migrations between regions due to the lockdown measure [25], our results suggest a consistent mapping between virtual network distance and physical distance during the pandemic. In other words, social interactions and emotional connections maintained through virtual platforms have mirrored geographical proximity despite restrictions on physical mobility. With limited data, we reveal that the correlation is not statistically significant for December 2019 (see Appendix Figure 7). This potentially suggests that this global health crisis not only impacts public health but also alters patterns of social emotion dynamics.

## 4.2 Percolation-based analysis of emotional resonance

By determining the percolation threshold $q_c$ of each emotional resonance network, we can identify the critical point at which the emotional resonance phenomenon becomes widespread and pervasive throughout the whole network (country), i.e., the emotional resonance index (ERI). As shown in Figure 5 (a), a clear phase transition point can be identified by tracking the $SG$ of the network for each month.

Given the relatively small scale of the network and the use of real-world Weibo data for weight calculation, the phase transitions observed here are not as sharp as those typically seen in a first-order phase transition. We also observe multiple $SG$ peaks in August and November. For simplicity, we select the first peak of $SG$ for all months when identifying the critical point. Figure 5 (b) presents the monthly ERIs. Consistent with our previous finding, January persists in an overall higher emotional resonance level with $q_c = 0.97$. As indicated by dash lines in the figure, three local peaks of the ERI can be observed. Indeed, the emotional resonance level is event-boosted. Besides January, the relatively higher emotional resonance level in April can be attributed to the Qingming Festival.

April 4th was assigned as a national mourning day to honor the victims of the pandemic. Given the increasing number of deaths during the pandemic, this festival particularly triggered emotional resonance among the population. The peak in December reflects people's surging emotions as they welcomed the new year with mixed feelings, encompassing both hope and anxiety under the ongoing pandemic.

As a reflection of human mental status, emotional resonance represents one of the primary channels through which people interact with and understand the external environment. The difference in emotional performance between developed and less developed cities stimulates us to examine the depth of reason that drives the way people share their expressed emotions via social networks. Several studies have indicated a potential correlation between public sentiment and the economic environment [8, 45]. To validate this hypothesis in the context of emotional resonance, we collect economic data from the National Bureau of Statistics of China (https://data.stats.gov.cn/index.htm). We investigate the correlation between the monthly ERI and the Total Retail Sales of Consumer Goods (TRSCG), which is the main indicator for the general economic environment. As indicated in Figure 5 (b), we found that ERI is negatively correlated with the month-on-month growth rate of TRSCG with a Pearson correlation coefficient $-0.69323$ ($p = 0.01243$, which is statistically significant at $p < 0.05$).

We check the correlation between ERI and other economic indicators (see Appendix Table 2), and all results indicate a negative relationship. Our analysis suggests that the pandemic greatly impacted both expressed emotions and economic conditions, and these two affected factors are interrelated. It is possible to monitor collective emotional resonance levels as an indicator of the country's economic health during global health shocks. We also compare the proposed percolation-based index ERI with the average sentiment score (see Appendix Figure 8 and Table 3). The monthly average sentiment scores are almost identical with slight variation ($0.85 \pm 0.02$), making them of limited significance for the heterogeneous study of emotion evolution.

Moreover, the network at percolation criticality has a very dilute structure and behaves as the "backbone" of the original network, which can be applied to identify bottlenecks as those observed in traffic networks [21]. In the case of an emotional resonance network, the bottlenecks are key pathways through which emotional resonance can spread. We identify the bottlenecks by comparing the remaining network just below and immediately above the percolation threshold. Figure 9 in the Appendix illustrates the removed links (in red) at $q_c$, showing that they can disintegrate the giant cluster and result in a maximal second-largest cluster. For example, the resonance between Henan and Shanxi is the critical pathway in January, and Shanxi ranks the highest frequency of connecting bottleneck links throughout the year. This suggests that Shanxi plays a pivotal role in stimulating emotional resonance, acting as a crucial bridge within the network during this period.

## 4.3 The temporal dimension of emotional resonance

Based on the definition of time-delayed cross-correlation, Figure 6 (a) illustrates the distribution of high-degree centrality nodes within

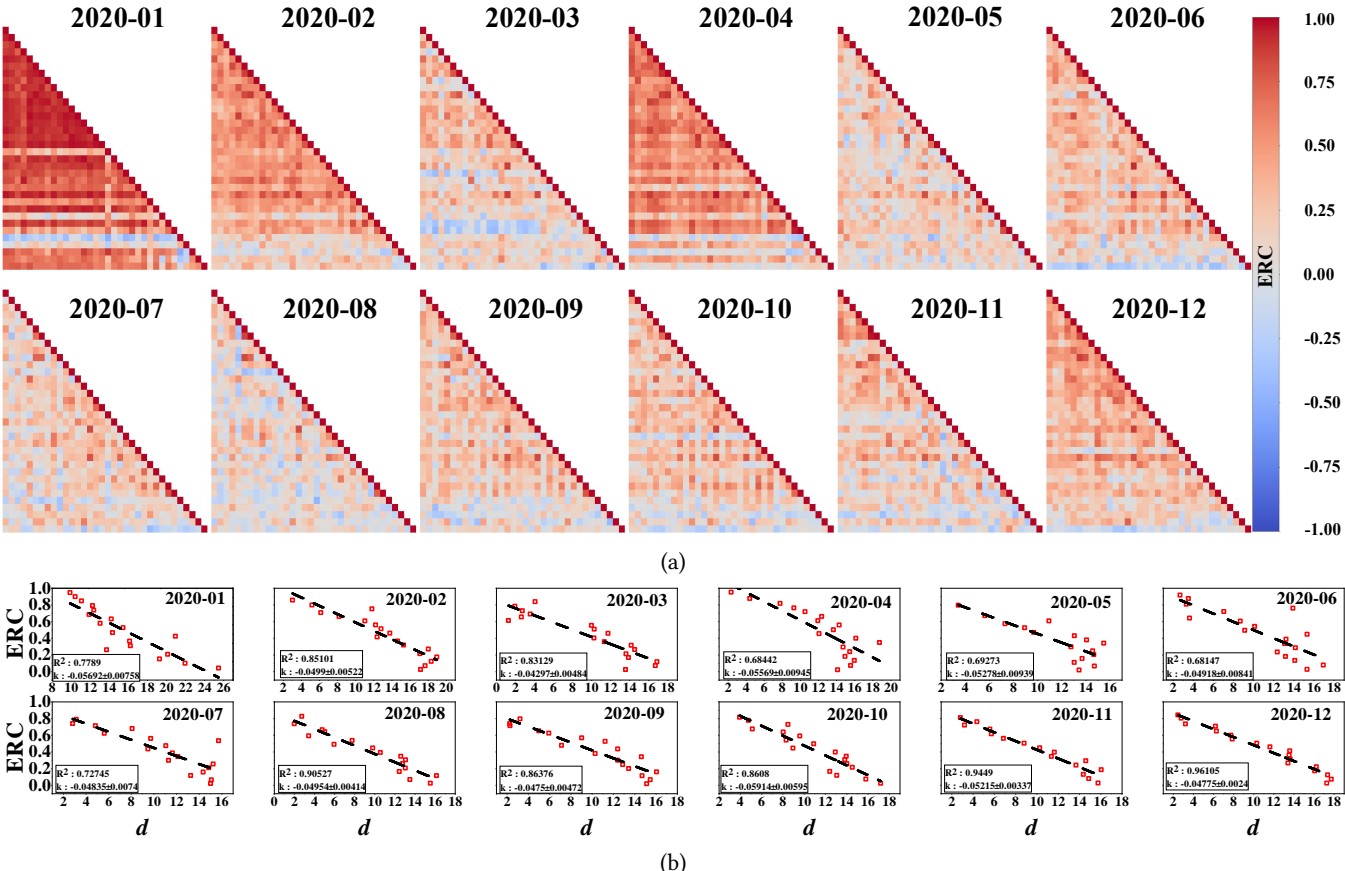

Figure 4: The spatial dimension of emotional resonance. (a) Heatmaps of the emotional resonance coefficients (ERC) between provinces through the year 2020. The horizontal (from top to bottom) and vertical (from left to right) axes are labeled with the following provinces in order: Hubei, Jiangxi, Hunan, Henan, Jiangsu, Zhejiang, Fujian, Shaanxi, Shanghai, Shandong, Chongqing, Shanxi, Guangdong, Sichuan, Tianjin, Guangxi, Beijing, Ningxia, Gansu, Hainan, Inner Mongolia, Yunnan, Liaoning, Qinghai, Jilin, Heilongjiang, Xinjiang, Macao, Hong Kong, and Taiwan. (b) ERC as a function of physical distance $d$ between each pair of provinces.

active tweeting regions (blue) and inactive regions (red). Our results reveal that these strong leaders and followers are primarily concentrated in inactive regions rather than active ones. Here, larger weighted outdegree values reflect a stronger leader effect of the node, while larger indegree indicates a stronger follower effect in terms of emotional resonance fluctuation. This indicates that leadership and followership in emotional resonance fluctuations do not necessarily depend on high levels of social media engagement and that inactive regions tend to play a dual role in influencing and reflecting online emotional responses to global health shock events.

The reveal of this dual-role characteristic of inactive regions on the social platform is of great importance. These regions not only serve as strong influencers in initiating significant emotional trends but also act as followers, responding sensitively to changes originating from other areas. In other words, the seemingly unimportant nodes, often found in regions with less social media engagement, have a significant impact on emotional transmission on social networks. By intervening in the emotional resonance of these key

nodes, we can mitigate their emotional effects on social networks. Therefore, understanding influential nodes in inactive areas can help develop effective strategies for disseminating public opinion or crisis management.

## 5 Discussion

Our analysis focuses on the collective emotional resonance under the most recent global health crisis, highlighting several key findings associated with how collectively expressed emotion propagates through social media platforms during the first year of the pandemic. Leveraging large-scale social media data and advanced sentiment analysis tools, we introduce a novel definition of emotional resonance in response to this unprecedented global health crisis. Despite occurring on virtual platforms, our findings reveal that emotional resonance diminishes with physical distance, extending previous research on the influence of geographic distance on online social interactions [20]. By applying percolation theory and

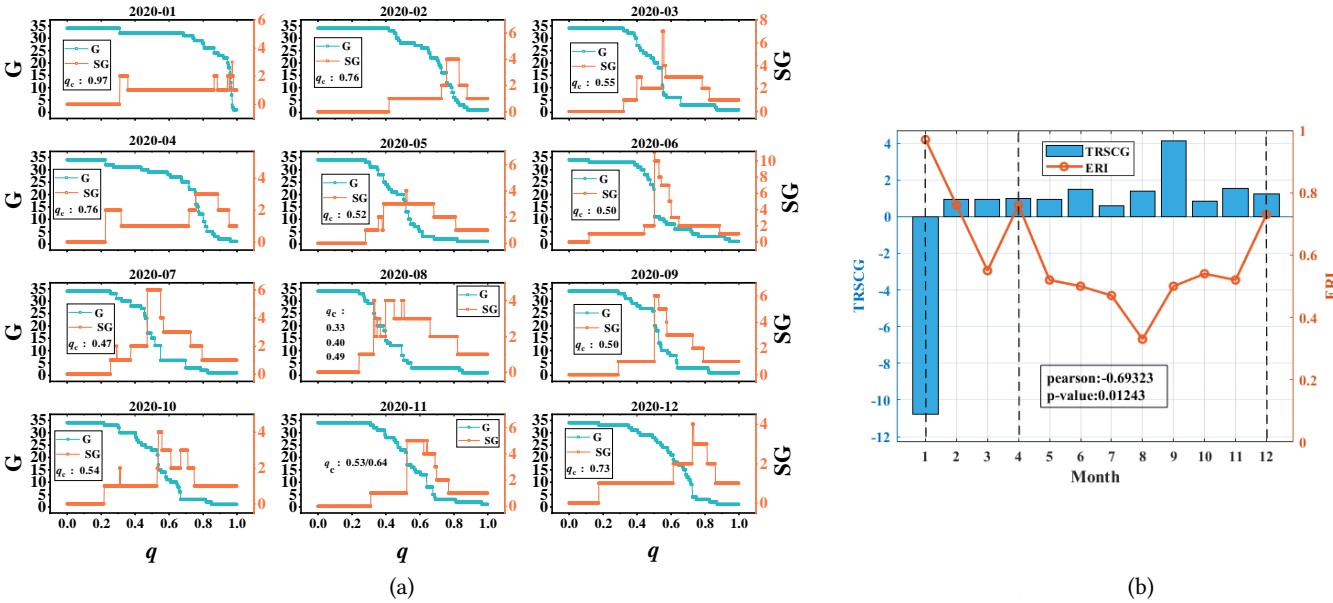

(a)   (b)

**Figure 5: Identification of the monthly emotional resonance index (ERI). (a) The phase transition points correspond to the maximal $SG$ values. (b) The monthly ERI was found to be negatively related to economic indicators.**

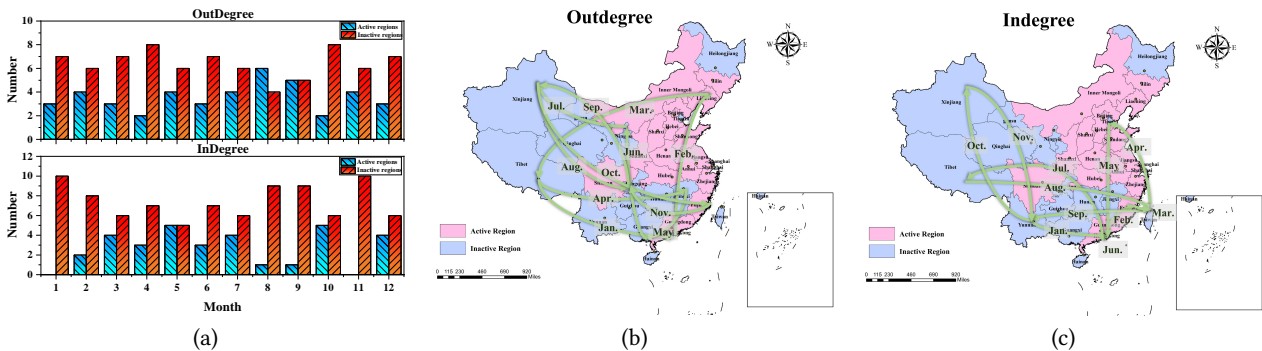

(a)   (b)   (c)

**Figure 6: Analysis of temporal dimension of emotional resonance based on time-delayed cross-correlation. (a) The distribution of high-degree centrality nodes (top 10) in active tweeting regions and inactive tweeting regions. The temporal shifting of nodes with the highest (b) outdegree and (c) indegree through the year.**

network analysis, we quantify the country-wide emotional resonance level and reveal the negative relationship between emotional resonance level and economic environment.

Our temporal analysis further captures the unique patterns of emotional resonance fluctuations across different regions and time periods. Based on the time-delayed cross-correlation, we reveal the dual-role characteristics of inactive tweeting regions. Our findings challenge conventional assumptions about the dominance of more active users [19, 30], which redefines the importance of users on social networks. In other words, more attention should be given to the 'edge users' when investigating the emotion propagation process since the social media engagement level alone does not fully account for their influence.

In summary, our study contributes methodologically by integrating sentiment analysis, spatial-temporal modeling, and percolation theory to study collective emotions comprehensively. In this post-pandemic era, our results provide a valuable framework for analyzing how collective emotional resonance propagates across social media platforms during such crises and its application to influential user identification, public opinion intervention, etc.

Although COVID-19 has passed, it will not be the last global pandemic we will face. Future research can enhance this framework by incorporating factors such as user demographics, media influence, and offline social interactions to deepen the understanding of emotional resonance. A larger-scale analysis of emotional resonance across global countries would also be of significant interest when data becomes available.

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

## A  Appendix

### A.1  Accuracy of the SEKP model

SKEP is designed for sentiment analysis to automatically identify and extract subjective information from text, including tendencies, positions, evaluations, and opinions. The Baidu research team further validated the effectiveness of the SKEP sentiment pre-training model across three typical sentiment analysis tasks: sentence-level sentiment classification, aspect-level sentiment classification, and opinion role labeling, using a total of 14 datasets in both Chinese and English. The experiments showed that initializing with the general pre-training model ERNIE (internal version), SKEP achieved an average improvement of about 1.2% over ERNIE and approximately 2% improvement over the previous state-of-the-art (SOTA). To assess the accuracy of the SKEP model in social media text emotion classification, we conducted additional validation using three social text datasets of different sizes.

The dataset contains time-stamped Weibo posts during the pandemic, with 2,000 labeled entries. Labels are 0 for neutral, 1 for positive, and 2 for negative (https://aistudio.baidu.com/datasetdetail/120950). Since the SKEP model is binary classification, we removed the neutral emotions from the dataset. After removal, the dataset still contains 1,189 entries. The accuracy is 82%.

The train_label.csv and train.csv datasets are sourced from the pandemic sentiment analysis of netizens' emotions (Pandemic Sentiment Analysis - PaddlePaddle AI Studio https://aistudio.baidu.com/aistudio/datasetdetail/24278/0). The train.csv file contains 100,000 data entries categorized as -1 (negative), 0 (neutral), and 1 (positive). The train_label.csv file contains 900,000 data entries with the same categories. During testing, we removed neutral entries and

identified binary classification results for negative and positive sentiments. The accuracy is shown in Table 1. After removing neutral emotions, train.csv contains 42,294 Weibo posts, and train_label.csv contains 366,813 Weibo posts.

It can be seen that the pre-trained SKEP model has high accuracy. These datasets, like the ones used in our paper, contain posts from Weibo users during the COVID-19 pandemic. We found that the SKEP model performs better with larger datasets. The excellent performance on these annotated datasets demonstrates that the SKEP model is also effective for our dataset.

**Table 1: Accuracy of SKEP**

| Model | Dataset | | |
|---|---|---|---|
| | Train.csv | Train_label.csv | label2000.json |
| SKEP_ernie_1.0_large_ch | 82% | 85.9% | 91.2% |

### A.2  Emotional resonance coefficient in December 2019

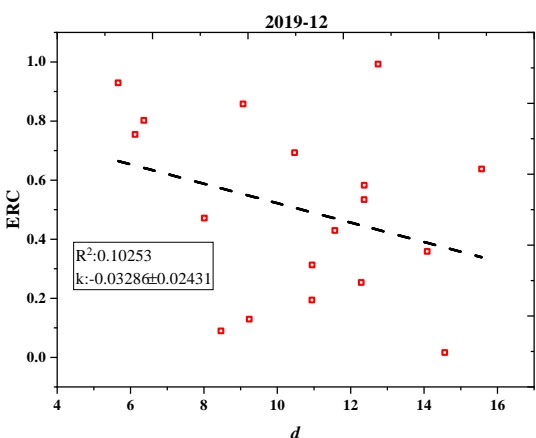

**Figure 7: Emotional resonance coefficient (ERC) as a function of physical distance *d* between each pair of provinces in December 2019. The correlation is not statistically significant.**

### A.3  Comparison between ERI and other economic indicators

The correlation between ERI and other economic indicators is shown in Table 2, and all results indicate a negative relationship. Here are the explanations of each indicator:

Percentages and Money Supply (M0, M1, M2) quantify money circulation at different levels. The concepts of money and quasi-money (M2) supply are important metrics in macroeconomics for measuring the total amount of currency in circulation within a country or region over a specific period (typically a month). M1: The most liquid forms of money, including all cash and demand deposits used for transactions. M0 refers to the currency in circulation.

Gold reserves refer to the gold held by a country's monetary authority, used to balance international payments, maintain or influence exchange rate levels, and serve as a financial asset.

Credit loans: When a borrowing enterprise obtains a loan from a bank for the first time, it's termed as a new loan. If the loan is repaid on time upon maturity and the enterprise applies for another loan afterward, this is also considered a new loan. However, if the enterprise fails to repay the loan on time and the bank agrees for the enterprise to borrow a second loan to repay the first, this is termed as "borrowing new to repay old." The year-on-year growth rate of loans is used here.

Chinese currency deposits in both RMB and foreign currencies: RMB is the domestic currency, while currencies from other countries are considered foreign currencies. The cumulative amount of these currencies deposited in domestic financial institutions is collectively referred to as Chinese currency deposits in both RMB and foreign currencies, used for comparative purposes.

Foreign Exchange Loans are loans issued by banks to enterprises using foreign currency as the unit of account. There are broad and narrow definitions of foreign exchange loans. Narrowly defined, foreign exchange loans refer specifically to loans issued by Chinese banks using foreign exchange funds absorbed from domestic enterprises and individuals, and lent to domestic enterprises. These loans are compared using cumulative amounts.

All data can be obtained from the National Bureau of Statistics (https://www.stats.gov.cn/) and Finance Data (https://data.eastmoney.com/).

**Table 2: Correlation analysis between ERI and economic indicators**

| Variable | Pearson Correlation | p-value |
| --- | --- | --- |
| TRSCG | -0.69323 | 0.01243 |
| M2 | -0.68374 | 0.01422 |
| M1 | -0.70388 | 0.01063 |
| M0 | -0.49435 | 0.10231 |
| Gold reserves | -0.65690 | 0.02030 |
| New stock investors | -0.58399 | 0.04618 |
| Domestic and foreign currency deposits in China | -0.63460 | 0.02665 |
| Foreign exchange loan data in China | -0.71586 | 0.00884 |
| Credit loans | -0.70513 | 0.01043 |

**Note: Pearson correlation coefficient, p-value < 0.05, is marked in bold.**

## A.4 Comparison ERI and the average sentiment scores

Observing Table 3 and Figure 8, we find that the Pearson correlation between the monthly average sentiment scores and most economic indicators shows no statistically significant results. This is because the monthly average sentiment scores are almost identical with little variation, making them of limited significance for the study. To capture the phenomenon of collective emotional evolution, using the average sentiment value alone is insufficient, as the values show little variation and lack meaningful measurement. Therefore, we use the Emotional Resonance Index (ERI) to measure emotional

resonance. This method allows us to assess it from a global perspective and better illustrate the phenomenon of emotional resonance, making it more valuable than using average sentiment values.

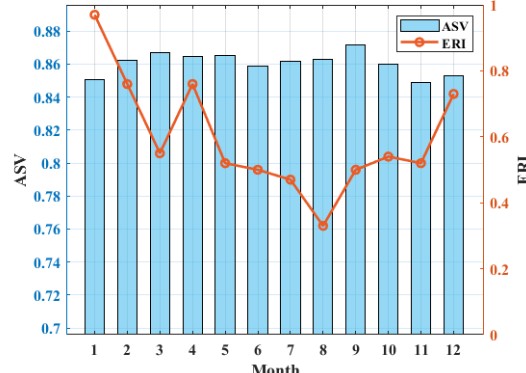

**Figure 8: Comparison between monthly ERI and average sentiment scores (denoted as ASV in the plot).**

**Table 3: Correlation analysis between average sentiment scores and economic indicators.**

| Variable | Pearson Correlation | p-value |
| --- | --- | --- |
| TRSCG | 0.52388 | 0.08042 |
| M2 | 0.37748 | 0.22640 |
| M1 | 0.06309 | 0.84557 |
| M0 | **0.59657** | **0.04060** |
| Gold reserves | 0.11387 | 0.72456 |
| New stock investors | 0.26913 | 0.39763 |
| Domestic and foreign currency deposits in China | -0.09116 | 0.77812 |
| Foreign exchange loan data in China | 0.01219 | 0.97002 |
| Credit loans | 0.29600 | 0.35023 |

**Note: Pearson correlation coefficient, p-value < 0.05, is marked in bold.**

## A.5 Identification of bottlenecks in the percolation process

The identification of bottleneck edges in the emotional resonance network for each month is illustrated in Figure 9. By comparing the network structure before and after the percolation threshold, the bottleneck links that are important to country-wide emotional resonance spread are colored in red.

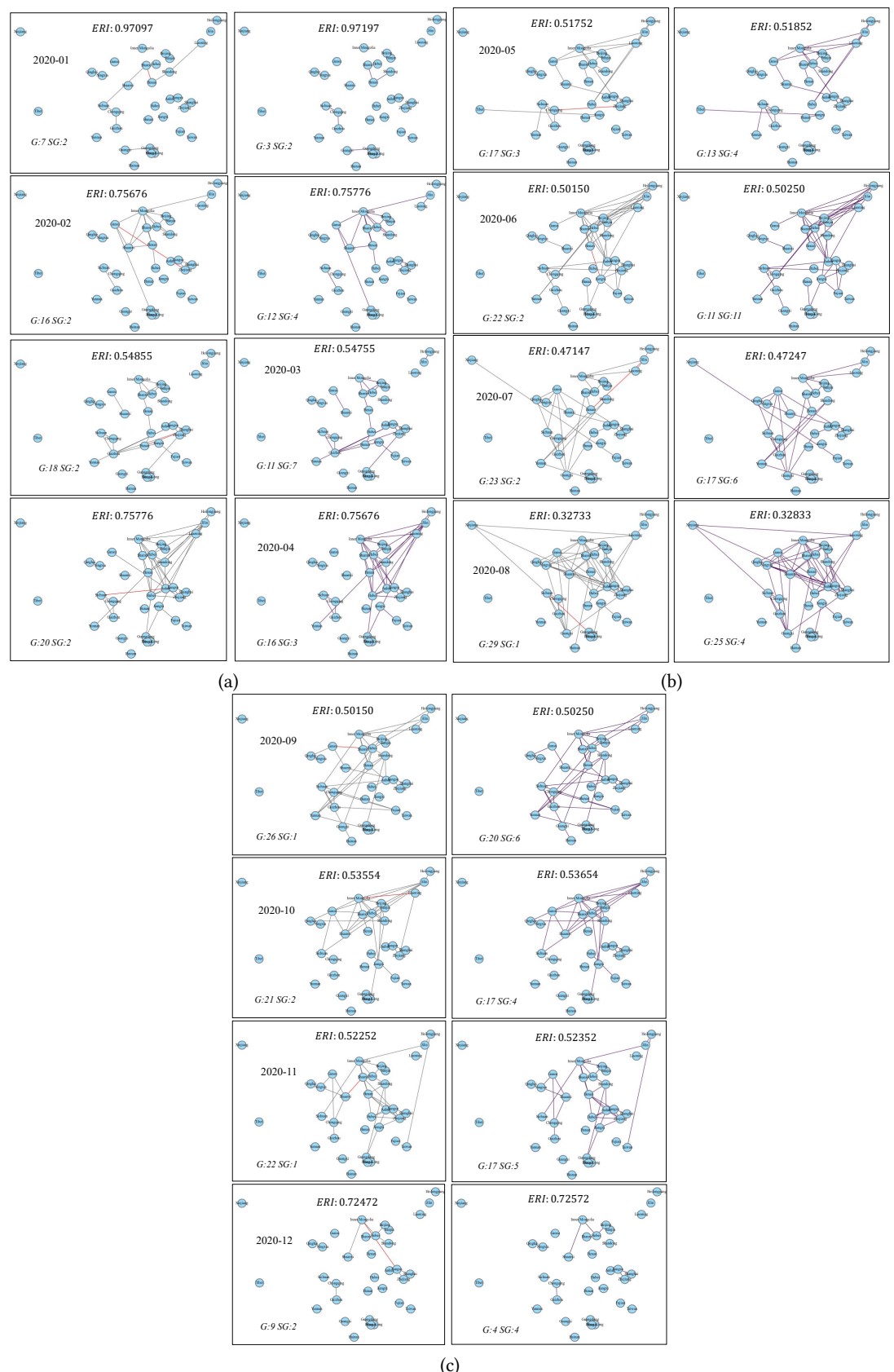

Figure 9: Visualizing the identification of bottlenecks in emotional resonance spreading. The bottlenecks are colored with red.