# OpenReview forum: "Spatial-temporal Analysis of Collective Emotional Resonance During Global Health Crisis"
_ACM.org/TheWebConf/2025/Conference — WWW 2025 Oral_

### Official Review · Reviewer_WxNz · 2024-11-24

**Novelty:** 3
**Technical Quality:** 2

**Review:**

This paper examines collective emotional dynamics on Sina Weibo during the initial year of the COVID-19 pandemic, with particular attention to the epidemic epicenter. The authors employ the Knowledge Enhanced Pre-training (SKEP) model to quantitatively measure tweet emotions and use Pearson correlation coefficients between provincial average sentiment scores to demonstrate that online emotional resonance correlates positively with geographic proximity. By redefining the critical parameter $q_c$ from percolation theory as a country-wide emotional resonance indicator (ERI), the authors establish negative correlations between economic indicators and ERI. The paper also identifies leader-follower patterns in emotional resonance fluctuations through time-lag emotion correlations.

Key Limitations:

1. Conceptual Ambiguity:

The paper lacks clear definitions of fundamental concepts, particularly "emotional resonance." The authors equate it with correlation between averaged sentiment scores, using this provincial-level metric to derive both country-level and geographic insights.

2. Methodological Concerns:

- The unidimensional nature of sentiment scores (positive/negative) provides limited insight. The authors' operational definition of "emotional resonance" may oversimplify the complex nature of emotional expression on social media, where users typically express nuanced viewpoints rather than binary sentiments.
- The provincial-based tweet categorization, while straightforward, overlooks critical factors such as population size/density differences and inter-provincial population movement.

3. Network Analysis Limitations:

The time-lag correlation analysis used to establish leader-follower patterns between provinces may not capture the essence of social media dynamics. Social networks fundamentally transcend geographical and temporal boundaries, suggesting that:
- User-to-follower emotional propagation patterns might be more relevant than province-to-province analysis.
- Geographically significant events (e.g., lockdowns) may influence emotion spread between users without direct social connections but geographic proximity, independent of provincial boundaries.

**Questions:**

Another major concern is that technical contributions in this paper is limited. Could the authors summarize the technique contributions?

**Reviewer Confidence:**

2: The reviewer is willing to defend the evaluation, but it is likely that the reviewer did not understand parts of the paper

**Scope:**

3: The work is somewhat relevant to the Web and to the track, and is of narrow interest to a sub-community

---

### Official Review · Reviewer_ck4v · 2024-11-28

**Novelty:** 5
**Technical Quality:** 5

**Review:**

The study focuses on a comprehensive analysis of the collective emotional response to the COIVD-19 pandemic. It aims to assess the emotional dynamics of the Chinese population during the first year of the pandemic, with a particular emphasis on the connection between the dynamics and geographical locations within China. The original dataset, which plays a pivotal role in the study, covers the Chinese population and Weibo tweets, posted between December 2019 and December 2020.

The analysis is based on three key indicators: (i) the developed emotional resonance coefficient (ERC), which is the Pearson correlation coefficient between the average sentiment scores for two provinces; (ii) a country-wide emotional resonance index (ERI), which represents a threshold of an ERC that results in network desintegration; and (iii) time-delayed cross-correlation, which quantifies the strength of a linear relationship between emotional resonance in one province and that in the other, but lagged.

The study reveals a positive correlation between online emotional resonance and geographic proximity, as well as the impact of online emotional resonance on various economic indicators. Additionally, it uncovers a leader-follower pattern in emotional resonance fluctuations.

The results are however location- and subject-specific, illustrating a number of effects that emerged during the first year of the pandemic. It is not yet clear whether these results will be applicable in similar situations in the future or in other regions with different cultural characteristics. In light of these limitations, the results of the study have not yet reached their full potential.

The following are comments/questions regarding the present study.

**Questions:**

1. What makes analyzing the Chinese population in the first year of the pandemic particularly noteworthy, despite the fact that China was the first country affected by the virus? Given that the study focuses on the impact of the pandemic through the lens of an online platform, it would be beneficial to consider the emotional status in other countries and regions as well.

2. The appendix includes a number of key methodological steps, although the main body of the article must be self-sufficient. Without a full review of the appendix, it is not possible to gain a complete understanding of the methodology, for example regarding the use of the SKEP model, calculation of sentiment scores, and the exclusion of ERC data.

3. The proposed conversion of negative emotional values into positive ones does not appear to be convincing. In this regard, two values can have completely opposite emotional connotations. I believe the author(s) may be overlooking a significant portion of the information here.

4. The analysis suggests that expressed emotions and economic conditions are interrelated. Certainly correlation does not imply causation, but could this be a result of the lockdown and the economic slowdown?

5. The paper examines the significance of the Pearson correlation coefficient. Are there any preconditions for the applicability of the statistical test in the data?

6. The rationale for setting the maximum lag to six days is not motivated.

**Reviewer Confidence:**

3: The reviewer is confident but not certain that the evaluation is correct

**Scope:**

3: The work is somewhat relevant to the Web and to the track, and is of narrow interest to a sub-community

---

### Official Review · Reviewer_aSov · 2024-12-02

**Novelty:** 2
**Technical Quality:** 3

**Review:**

The focus on collective emotional resonance during a global health crisis is relevant. However, the lack of clear articulation of methods and findings diminishes the perceived novelty and impact of the work.

## Strengths
- The paper's relevance is highlighted as its focus on collective emotional resonance during a global health crisis is significant.
- The spatial-temporal approach has the potential for original insights if properly refined.

## Weaknesses

### Writing and Terminology
- The writing is unclear and contains numerous typos, making it difficult to follow.
- The inconsistent terminology, with references to both "tweets" and "posts," creates confusion.

### Dataset and Methodology
Key elements such as the dataset and analysis methodology lack adequate explanation:
- The information about the structure of the data is missing. For instance, the authors do not discuss whether all the posts are geotagged and, if so, how the geotag is provided (e.g., GPS, self-written) and what are the potential implications/biases.
- Poor explanation of key calculations (e.g., monthly ERC, range of ERC values).
- Lack of explainability regarding some theories discussed (e.g., Percolation theory, on page 4) and definitions (e.g., what is an active region?).

### Structure
- The structure could be improved by moving some of the results presented prematurely in the Introduction to dedicated sections like *Data and Methods*.

### Visualizations
- Figures and captions, especially Figure 1, require significant improvement to enhance interpretability. For example, events corresponding to spikes in post counts should be labeled to provide meaningful context.

### Appendix
- Improve the explanation in the Appendix, including naming labels on Table 1 in a more intuitive way.

### Additional Questions
1. **Effect of Posts per Region:**
   - What is the effect of the number of posts per region in the average? Is it possible to normalize the posts by the total number of overall posts that come from a region?
     - This normalization could help control for disparities in activity levels across regions and mitigate biases from over-represented areas.
     - For example:
       - Use proportions or weighted averages based on regional post activity.
       - Include this normalization step in the *Methods* section with detailed reasoning.

2. **M₀ in Tables 2 and 3 (Appendix):**
   - M₀ is the only economic indicator not statistically significant in Table 2 but the only statistically significant one in Table 3.
   - What hypotheses or explanations account for this inconsistency?

## Final Recommendation Summary
- Revise and proofread the text for clarity and error correction.
- Use consistent terminology throughout the paper.
- Provide a comprehensive dataset description, including geotagging details, language, and potential biases.
- Reorganize content to ensure results appear in appropriate sections, with adequate explanation of methods and theories.
- Improve figures with annotations and add context to facilitate interpretation.
- Consider normalizing metrics by post volume to enhance the robustness of regional comparisons.

**Questions:**

See my comments above.

**Reviewer Confidence:**

3: The reviewer is confident but not certain that the evaluation is correct

**Scope:**

3: The work is somewhat relevant to the Web and to the track, and is of narrow interest to a sub-community

---

### Official Review · Reviewer_idzG · 2024-12-02

**Novelty:** 5
**Technical Quality:** 5

**Review:**

The authors have analyzed an extensive number of geotagged tweets from Sina Weibo to study collective emotional dynamics among the Chinese population during the initial year of the Covid-19 pandemic, using several computational social science methods in an innovative way. As far as I know, similar data collection and analysis have not been done before.

In general, the article was easy to follow and well written. There were a couple typos: 3.1 resona -> resonance and 3.3 the network undergoes -> the network undergoing. I also suggest that the method section is written in the past tense. In addition, the research questions or hypotheses to be answered in the article could be more clearly stated in the beginning.

The study revealed a positive relationship between online emotional resonance and geographic proximity and identified a leader-follower pattern in emotional fluctuations. The proposed implications include online interventions during global health crises.

The article did not discuss limitations, such as the interpretation of sentiment, demographic differences between regions, or other biases in the material. A simple dichotomous sentiment analysis does not allow an in-depth analysis of the social significance of the emotions caused by the pandemic. In order to understand the real-life implications, it would be important to know where the emotions stem from, for example, health concerns, stress caused by lockdown, opposition to restrictions, vaccine criticism? Particularly if this approach would be used for mental health monitoring and potential targeted interventions, it would be important to minimize the risk of misinterpretations. In addition, the article did not address ethical concerns and risks, such as possible adversarial attacks.

Pros
The article aims to produce new information from extensive social media data by innovatively combining various methods.
The research is well reported and illustrated with figures.
The list of sources is extensive and detailed documentation of the data and methods used has been made available.

Cons
The article did not discuss ethical concerns or limitations, such as the interpretation of sentiment, demographic differences between regions, or other biases contained in the material.

**Questions:**

- Would a better understanding of the special nature of emotions during the pandemic be obtained, for example, by comparing the results to the time before and/or after the pandemic?
- Section 4.2 “The difference in emotional performance between developed and less developed cities stimulates us to examine the depth of reason that drives the way people share their expressed emotions via social networks.” Could you please clarify what “emotional performance” and “depth of reason” mean in this sentence?
- What are the most important limitations of the research and could they be discussed in the text? For example:
What about the role of trolling, advertising, disinformation etc.?
What are the demographics and percentage of people using the service, i.e. how generalizable are the results to the population?
What are the demographic characteristics, and to what extent do they explain the results?

**Reviewer Confidence:**

2: The reviewer is willing to defend the evaluation, but it is likely that the reviewer did not understand parts of the paper

**Scope:**

4: The work is relevant to the Web and to the track, and is of broad interest to the community

---

### Official Review · Reviewer_7ZPQ · 2024-12-03

**Novelty:** 4
**Technical Quality:** 5

**Review:**

This work proposes several concepts to incorporate geographic proximity and temporal evolution into the analysis of emotional resonance. The authors conducted a case study using a unique Chinese social media dataset from the first year after the onset of COVID-19. The study presents some interesting findings, such as a negative correlation between economic indicators and collective emotional resonance, and the observation that strong leaders and followers tend to concentrate in inactive regions rather than active ones. Overall, the topic is promising and has significant social value. However, there are some methodological and writing issues that need to be addressed to strengthen the paper.

Pros:

1. Impactful topic
2. Thoughtful mathematic design
3. Interdisciplinary work linking advanced techniques to social theories

Cons:

1. Assumption is weak
2. Interpretation of results is kind of shallow
3. Overall writing needs improvement in terms of storytelling.

**Questions:**

1. Comprehensiveness of Emotional Resonance Analysis
   The analysis focuses on the absolute value of emotional strength when studying emotional resonance and excludes negative correlations. This approach raises two concerns that may affect the comprehensiveness of the analysis: (1) Positive and negative emotions can interact and mitigate each other when one region is influenced by another. This interaction is lost when using absolute values. (2) Clear positive and negative correlations of actual emotions between regions are crucial for producing meaningful results in social analysis. Would introducing a relative emotion metric (e.g., based on the deviation from the average emotion for each region) provide a more nuanced alternative to simply dropping the emotional sign? This adjustment could offer a clearer understanding of inter-regional influences and their directionality.
2. Choice of q-Threshold for Network Development (Figure 3\)
   The values for the q-threshold used to construct the province network in Figure 3 appear arbitrary. How were these values determined? Providing a rationale or sensitivity analysis for the choice of these thresholds would enhance the credibility of the results.
3. Placement of Technical Details in the Introduction
   The last paragraph of the introduction delves into technical details that may disrupt the flow of the introduction. Moving these details to the methodology section could improve the paper's structure and readability.
4. Choice of Economic Indicator
   In Section 4.2, the correlation between economic status and emotion is analyzed using the Total Retail Sales of Consumer Goods (TRSCG) as the economic indicator. While this is a valid measure, commonly used indicators in economic studies include total GDP or average GDP per capita. Have the authors considered these alternatives? A comparison of results using different economic indicators could provide a broader perspective on the findings.
5. Temporal Analysis of COVID-19 Events
   Since this study discusses emotional resonance during the COVID-19 pandemic, incorporating a temporal analysis of significant COVID-19-related events (e.g., lockdowns, policy changes) could make the work more compelling. This could help contextualize the emotional evolution over time and provide deeper insights into the social impacts of the pandemic.
6. Figure Captions and Readability
   Generally, the figure captions could be made larger and more detailed to improve readability and accessibility.

**Reviewer Confidence:**

4: The reviewer is certain that the evaluation is correct and very familiar with the relevant literature

**Scope:**

3: The work is somewhat relevant to the Web and to the track, and is of narrow interest to a sub-community